# Core-Shell Structured SiO_2_@NiFe LDH Composite for Broadband Electromagnetic Wave Absorption

**DOI:** 10.3390/ijms24010504

**Published:** 2022-12-28

**Authors:** Zhilan Du, Dashuang Wang, Xinfang Zhang, Zhiyu Yi, Jihai Tang, Pingan Yang, Rui Cai, Shuang Yi, Jinsong Rao, Yuxin Zhang

**Affiliations:** 1College of Material Science and Engineering, Chongqing University, Chongqing 400044, China; 2No. 59 Research Institute of China Ordnance Industries, Chongqing 400039, China; 3School of Automation, Chongqing University of Posts and Telecommunications, Chongqing 400065, China

**Keywords:** SiO_2_ microspheres, LDH, microwave absorption, core-shell structure

## Abstract

In this work, a novel core-shell structure material, NiFe layered double hydroxide (NiFe LDH) loaded on SiO_2_ microspheres (SiO_2_@NiFe LDH), was synthesized by a one-step hydrothermal method, and the spontaneous electrostatic self-assembly process. The morphology, structure, and microwave absorption properties of SiO_2_@NiFe LDH nanocomposites with different NiFe element ratios were systematically investigated. The results show that the sample of SiO_2_@NiFe LDH-3 nanocomposite has excellent microwave absorption properties. It exhibits broadband effective absorption bandwidth (RL < −10 dB) of 8.24 GHz (from 9.76 GHz to 18.0 GHz) and the reflection loss is −53.78 dB at the matched thickness of 6.95 mm. It is expected that this SiO_2_@NiFe-LDH core-shell structural material can be used as a promising non-precious, metal-based material microwave absorber to eliminate electromagnetic wave contamination.

## 1. Introduction

In recent decades, with the wide application of electronic equipment in military, medical, life, and other scenarios, as well as the rapid development of 5G technology, electromagnetic interference and electromagnetic radiation problems have become increasingly serious, thereby presenting an urgent demand for electromagnetic absorption (EMA) materials [1,2,3,4]. It is well known that high-performance microwave absorbers working in the S, C, and X bands have achieved certain research progress [5]. As a promising absorber, SiO_2_ not only has the advantages of low density, high stability, good dispersion, and structure, but also has excellent biocompatibility and can be combined with one-dimensional nanowires [6], two-dimensional nanosheets [7,8,9], effectively combining related novel structures [10,11,12]. In recent years, researchers believe that designing materials with core-shell structures are expected to make them effective microwave-absorbing materials. Wang et al. successfully prepared material graphene@SiO_2_@NiO nanoflowers, which have good microwave absorption performance, and their RLmin value is −20.5 dB [13]. Qiao et al. constructed a novel egg yolk shell Fe_3_O_4_@void@SiO_2_@PPy nanochains with a minimum RL value of −54.2 dB and an effective absorption bandwidth (RL < −10 dB) of 5.90 GHz [14]. Huang et al. successfully prepared a gradient FeNi-SiO_2_ film to achieve a reflection loss of −55.2 dB at 2.5 mm [15]. These studies show that the microwave absorption performance of SiO_2_ depends to a large extent on the structure and properties of its Appendix A [16]. Therefore, it is very meaningful to search for novel structural SiO_2_-based core-shell materials with enhanced microwave attenuation properties.

Layered double hydroxides (LDHs) are known as 2D hydrotalcite-like materials and consist of a positively charged host layer and exchangeable interlayer anions with the general formula $[M_{1-x}{}^{2+}M_{x}{}^{3+}{\left(-\mathrm{OH}\right)}_{2}{\left(A^{n-}\right)}_{x/n}\cdot {\mathrm{mH}}_{2}O$], where M^2+^ and M^3+^ represent divalent and trivalent metals, and An- represent interlayer anion [17]. As the layered double hydroxides (LDHs) containing transition metal elements (Fe, Co, Ni, etc.) with high surface-to-volume ratio, controllable layered structure and tunable metal composition are expected to be excellent microwave absorbers [18]. LDH not only has a high specific surface area and a rich inner interface but also a stable point of structure, which provides favorable conditions for multiple reflections of incident electromagnetic waves after entering the material, thereby minimizing the attenuation of electromagnetic waves [19]. In addition, the relatively low conductivity of LDH can provide a good impedance match with the magnetic material composite so that more electromagnetic waves can easily enter the material, meeting the two advantages of high-quality absorbing materials. Therefore, LDH can be considered a promising electromagnetic wave absorbing material [20]. Moreover, compared with other LDHs, NiFe LDH is characterized by magnetic properties due to the presence of transition metals Ni and Fe, which can introduce a magnetic loss mechanism, and the magnetic-dielectric synergy can effectively match the characteristic impedance over a wide frequency range, thus broadening the effective absorption bandwidth [21]. Therefore, NiFe LDH is expected to be a promising material for electromagnetic wave absorption.

Currently, excellent electromagnetic wave absorption performance can be achieved by designing a core-shell structure of absorbers that combine dielectric loss and magnetic loss components. Well-designed core-shell nanosheets have a large specific surface area and abundant interfaces. With SiO_2_ as the core, it is necessary to select a material with different loss mechanisms as the shell, meet the impedance matching conditions, and the different components work synergistically to improve the electromagnetic wave absorption performance [22]. Studies have shown that core-shell materials that satisfy both dielectric loss and magnetic loss have better performance and application prospects [23]. Magnetic loss, dielectric loss, dipole polarization, and superior impedance matching work together to obtain a microwave absorber with lightweight, thin thickness, wide frequency band, and strong absorption [24]. It has been found that low-dimensional, core-shell materials have considerable potential in improving the absorption performance of electromagnetic waves through their special structural properties and combined with the interface effect of the material itself, magnetic loss, dielectric loss, and impedance matching [25]. On this basis, magnetic LDH is very suitable as one of the candidate shell materials on SiO_2_ due to its environmental stability, excellent magnetic permeability, and novel bilayer structure [26]. However, there are very few related reports so far. We can improve the impedance matching of SiO_2_ materials by constructing LDH structures with different NiFe ratios, which endows a designable combination of spherical and layered structures, good matching of magnetic and electrical conductivity, and larger characteristic of the specific surface area, thereby enabling the material to obtain strong loss and broadband absorption properties at the same time [27]. By combining the excellent advantages of nano-SiO_2_ and NiFe LDH, not only can the magnetic loss of NiFe LDH be fully utilized, but also the harmful electromagnetic waves can be further dissipated through the interface interaction between SiO_2_ and NiFe LDH [28].

In this work, a one-step hydrothermal method was used to design and synthesize bimetallic hydroxide (NiFe-LDH) nanosheets grown on the surface of SiO_2_ microspheres. Magnetic NiFe-LDH has a complex three-dimensional structure. By adjusting the ratio of Ni^2+^ and Fe^3+^ in NiFe LDH, good impedance matching, and high-performance microwave absorption can be obtained. The prepared SiO_2_@NiFe LDH nanocomposites with different NiFe ratios have high impedance matching, the superior synergistic effect of dielectric loss and magnetic loss, and excellent microwave absorption performance. SiO_2_@NiFe LDH nanocomposites have the characteristics of strong microwave scattering ability, light weight, strong absorption performance, and wide absorption bandwidth and are expected to become a new type of structural microwave absorption material.

## 2. Results

### 2.1. Structure Characterization and Analysis

The fabrication process of SiO_2_@NiFe-LDH is displayed in Figure 1. Firstly, the prepared SiO_2_ microspheres were uniformly dispersed in an aqueous solution. A mixed solution containing nickel nitrate and ferric nitrate was added to the aqueous solution with urea a as precipitant. After the solution was mixed evenly by strong magnetic stirring, the solution was put into a rotary oven and heated at 180 °C for 24 h, and the brown precipitate was finally obtained. Under hydrothermal conditions, urea (CO(NH_2_)_2_) decomposes into ammonium ions (NH_4_^+^), carbon dioxide (CO_2_), and hydroxide ions (OH^−^). Then carbon dioxide (CO_2_) is hydrolyzed into carbonate ions (CO_3_^2−^) and hydrogen ions (H^+^). At the same time, metal cations (Ni^2+^ and Fe^3+^) combine with the hydroxide (OH^−^) to form layered double hydroxides on a silica substrate, while carbonate ions (CO_3_^2−^) act as interlayer anions to maintain charge balance [29].

To characterize and analyze the morphologies and lattice of the as-prepared samples on the nanoscale, SEM and TEM images were acquired, as shown in Appendix A, and Figure 1. The TEM images of Appendix A and Figure 1a–c show that SiO_2_@NiFe-LDH has a core-shell nanostructure, and the thickness of the LDH shell layer is less than 100 nm. As can be seen from the TEM images, the outer layer of SiO_2_@NiFe-LDH-3 consists of a large number of nanosheets, which are superimposed together to form a large number of accumulation pores, forming a rich interface between each other. In Appendix A, when the molar ratio of Ni^2+^ and Fe^3+^ is 3:3, the nanosheets are thinner and grow vertically on the surface of SiO_2_ spheres, which is conducive to improving the charge transfer efficiency. According to the HAADF mapping image in Figure 1e–i, there are Si, O, Ni, and Fe elements in SiO_2_@NiFe LDH-3, SiO_2_ microspheres provide Si and O elements, and LDH provides Ni, Fe, and O elements.

The XRD spectra of Appendix A confirm the composition of pure SiO_2_. Wide peaks in the XRD spectrum of SiO_2_ indicate that it is an amorphous phase. Due to its inherent hydrophilicity, SiO_2_ microspheres are conducive to the adsorption of Fe^3+^ and Ni^2+^ ions, promote the uniform coverage of Fe^3+^ and Ni^2+^ on the surface of SiO_2_ microspheres, and grow in situ to form NiFe LDH nanosheets. In the XRD pattern (Figure 2a), the four diffraction peaks located at 2θ of 11.3°, 22.5°, 34.5°, and 59.7° correspond to (003), (006), (012) and (110) crystal planes, respectively, which are the characteristic diffraction peaks of LDH [30], further illustrating the successful in situ growth of a typical hydrotalcite-like material on the surface of SiO_2_, wherein (003) the crystal plane spacing is 0.68 nm.

The hysteresis loop (M-H curve) of the SiO_2_@NiFe LDH composites tested under 300 k conditions in the range of −15 Koe to 15 Koe is shown in Figure 2b. It can be seen that samples with three different molar ratios all exhibit typical weak ferromagnetic properties, that is, the magnetization intensity of the particles changes slightly with the increase of the applied magnetic field, and then reaches a saturation state. The saturation magnetization intensity Ms of SiO_2_@NiFe LDH-1,2,3 was 0.75, 1.15, and 1.40 emu/g, respectively. As the molar ratio of Fe^3+^ increases, the coercivity of SiO_2_@NiFe LDH composites also gradually increases. Among them, the S3 sample has the greatest coercive force (24.8 Oe).

XPS spectroscopy further verifies that the elements Ni, Fe, C, Si, and O in SiO_2_@NiFe-LDH-3 (Figure 2c) conform to HAADF results. The combined energy of 856.5 eV and 874.1 eV is attributed to Ni 2p_3/2_ and Ni 2p_1/2_, respectively, representing Ni^2+^ in SiO_2_@NiFe-LDH microspheres. Signal sum at 713.3, 726.6 eV corresponds to Fe 2p_3/2_ and Fe 2p_1/2_, implying Fe^3+^. XPS surveys of S3 (Figure 2b) showed that the six peaks were located at 857.23, 713.44, 533.7, 285.07, and 102.34 eV for Ni 2p, Fe 2p, O1s, C1s, and Si, where the corresponding atomic concentrations of Ni, Fe, O, C, and Si were 13.54%, 4.07%, 65.65%, 8.60%, and 8.14%, respectively.

FT-IR spectroscopy was performed on pure SiO_2_ microspheres and SiO_2_@NiFe LDH (Appendix A). The wide peak of pure SiO_2_ near 1011.97 cm^−1^ is due to the bending vibration of the physical absorption of water, and the weak adsorption peak at around 791 cm^−1^ is due to the existence of the Si-O-Si bond. The characteristic peaks of -OH and physical adsorption water still exist at about 3384.77 cm^−1^ and 1636.80 cm^−1^ of SiO_2_@NiFe LDH. The weak peak near 1007.78 cm^−1^ is considered to be the intercalation structure of -OH and CO^3−^ in hydrotalcite (NiFe LDH). The bending vibration mode of metal hydroxide octahedral complexes (Ni-O and Fe-O) is observed at 594.97 cm^−1^ and 663.99 cm^−1^, respectively. The results show that NiFe LDH is formed on SiO_2_ microspheres, which is consistent with the XRD characterization results. Thermal analysis (Appendix A) shows that SiO_2_@NiFe LDH-1, 2, 3 have two significant mass reduction processes during the temperature rise from 30 °C to 1000 °C. The weight loss of the S3 sample was 5.5635% at about 150 °C and 7.0129% at about 350 °C. This phenomenon is caused by the evaporation of adsorbed water and carbonate anion intercalation transition in NiFe LDH, respectively.

### 2.2. Electromagnetic Parameters and Microwave Absorption Properties

The composite material and paraffin were mixed evenly in a mass ratio of 4:6, and they were pressed into a circular sample using a mold with an inner diameter of 3.04 mm, an outer diameter of 7.00 mm, and a thickness of 2.00 mm. The composite permittivity ($\varepsilon_{r}=\varepsilon^{\prime }-j\varepsilon^{\prime \prime }$) and permeability ($\mu_{r}=\mu^{\prime }-j\mu^{\prime \prime }$) were obtained by the coaxial method with a vector network analyzer in the frequency range 2–18 GHz., The electromagnetic wave absorption performance depends on electromagnetic parameters of materials [31]. This section systematically investigates the electromagnetic parameters of the absorbers of the prepared samples combined with paraffin.

Figure 3 shows the relative permittivity and relative permeability of the absorber. It can be seen from Figure 3a that the $\varepsilon^{\prime }$ constant decreases gradually, and the value is between 3 and 4, which is lower than the corresponding value of other NiFe-based materials reported in the literature [32,33]. For wave absorbing materials, an excessively high dielectric constant will generate an oscillating current on the surface of the material, and electromagnetic waves will be reflected instead of absorbed. On the contrary, a low dielectric constant is conducive to impedance matching, that is, it can allow more electromagnetic waves to be incident on the surface of the material. Among the materials, it has great benefits for the absorbing properties [34]. The gradually decreasing dielectric constant can be attributed to the typical dispersion response due to the increasing hysteresis of the dipole polarization response with respect to the electric field [35]. The relative permittivity of absorbing materials mainly depends on the space charge polarization and dipole polarization in the material, which have a strong dependence on frequency. It can be seen from Figure 3b that the imaginary part of the permittivity has obvious fluctuations in the high frequency part (14–18 GHz), which means that there is obvious polarization loss in the decay process of the incident electromagnetic wave and is due to the electron exchange between the Fe^3+^ and the Ni^2+^ in the NiFe-LDH inducing a large number of space charges and polarization charges under the action of the alternating electromagnetic field. Both polarizations increase the dielectric loss capability of the absorber, which is consistent with the frequency-dependence conclusion of our previous analysis [36]. Usually, the real part ($\varepsilon^{\prime }$ and $\mu^{\prime }$) of the electromagnetic parameter represents the energy storage capacity of the absorbing material for electromagnetic energy, while the imaginary part ($\varepsilon^{\prime \prime }$ and $\mu^{\prime \prime }$) represents the dissipation capacity of electrical and magnetic energy, respectively [37]. The real part and imaginary part of the dielectric constant of SiO_2_@NiFe LDH-3 (S3) are the largest, which indicates that S3 has stronger electromagnetic storage ability and dielectric loss ability than the other two samples.

Figure 3d,e show the variation curves of $\mu^{\prime }$ and $\mu^{\prime \prime }$ of SiO_2_@NiFe LDH with frequency. The three samples generally show a downward trend, and the fluctuation is more obvious. There are many factors that affect the $\mu^{\prime }$ value, such as resonance, magnetic crystal anisotropy, exchange coupling effect and eddy current [38]. After the hydrothermal reaction, NiFe bimetals form a layered bimetallic structure on SiO_2_ microspheres, and with the increase of the relative content of Fe, which has strong magnetic properties, the saturation magnetization also corresponds. Enhancement (Figure 2b), the high Ms (1.5 emu/g), and low Hc (20 Oe) of the S3 samples help to improve the magnetic permeability and enhance the electromagnetic wave attenuation capability [39]. Therefore, the real part of the magnetic permeability of the S3 is relatively high. For the imaginary part curve, the three samples generally show a downward trend with obvious fluctuations. The trends of the three samples are very similar. Usually, the cause of magnetic loss is that the natural resonance behavior of the absorber causes large fluctuations in the low-frequency band, and the fluctuations in the high-frequency band are attributed to the switching resonance, which is part of the absorber’s absorption mechanism [40]. 

The dielectric loss tangent and the magnetic loss tangent represent the dielectric loss capability and magnetic loss capability of the material, respectively. The higher the tan *δ_ε_* or tan *δ_μ_*, the greater the ability of the sample to absorb electromagnetic waves through dielectric loss or magnetic loss [41]. The decreasing trend of tan *δ_ε_* of the three samples in Figure 4c is almost the same, but there is a large fluctuation from 14 to 18 GHz, showing several obvious resonance peaks, which are similar to the resonance peaks of $\varepsilon^{\prime \prime }$, indicating that there are multiple polarizations on the absorption surface relaxation, such as interfacial polarization and dipole polarization. The massive heterojunction between NiFe nanosheets on the microspheres and the SiO_2_ matrix produces interfacial polarization, which weakens incident electromagnetic waves and leads to dielectric loss. In addition, the nano-sized NiFe nanosheet layered structure can act as a dipole, generate a large amount of orientation polarization, and convert electromagnetic wave energy into thermal energy, especially at low frequencies, indicating that S3 exhibits the best dielectric at low frequencies loss capacity. The difference in tan *δ_μ_* values of the three samples in the figure is very small, and they all fluctuate between 0.02 and 0.24, indicating that the three samples have similar magnetic loss capabilities. In addition, the tan *δ_μ_* value of the three samples is greater than the tan *δ_ε_* value at low frequencies; and the tan *δ_ε_* is greater than the tan *δ_μ_* value at high frequencies, indicating that the microwave absorption properties of the three samples originate from dielectric loss and magnetic loss and depend on the magnetic field at low frequencies. Loss depends on dielectric loss at high frequencies.

Debye relaxation is one of the important ways in which absorbing materials can produce dielectric losses. Usually Equation (3) can be derived from Equations (1) and (2), which can express the relationship between $\varepsilon^{\prime }$ and $\varepsilon^{\prime \prime }$ [42].
(1)$$\varepsilon^{\prime }=\varepsilon_{\infty }+\frac{\varepsilon_{s}-\varepsilon_{\infty }}{1+\omega^{2}\tau^{2}}$$
(2)$$\varepsilon^{\prime \prime }=\frac{\varepsilon_{s}-\varepsilon_{\infty }}{1+\omega^{2}\tau^{2}}\omega \tau +\frac{\sigma }{\omega \varepsilon_{0}}$$

Which could also be written as:(3)$${\left(\varepsilon^{\prime }-\frac{\varepsilon_{s}-\varepsilon_{\infty }}{2}\right)}^{2}+{\left(\varepsilon^{\prime \prime }\right)}^{2}={\left(\frac{\varepsilon_{s}-\varepsilon_{\infty }}{2}\right)}^{2}$$
where $\varepsilon_{s}$ and $\varepsilon_{\infty }$ are the static permittivity, optical permittivity, vacuum permittivity and angular frequency, respectively. Therefore, each semicircular curve represents a relaxation process. The SiO_2_@NiFe LDHs exhibit obvious multiple Cole–Cole semicircles in Appendix A, but the semicircles have some disorder, which indicates strong multipole relaxation and decay, thereby confirming that SiO_2_@NiFe LDH may be a mixed polarization model with interfacial polarization, electric dipole polarization, etc.

The reflection loss value (RL) of the absorber at 2–18 GHz corresponding to different thickness frequencies is calculated using the transmission line theory [43].
(4)$$\mathrm{RL}\left(dB\right)=20\log\left|\left(Z_{in}-Z_{0}\right)/\left(Z_{in}+Z_{0}\right)\right|$$
(5)$$Z_{in}=Z_{0}\sqrt{\mu_{r}/\varepsilon_{r}}\tanh\left[j\left(2\pi fd/c\right)\sqrt{\mu_{r}\varepsilon_{r}}\right]$$
where c, f, and d are the speed of light, the corresponding frequency, and the matching thickness, respectively. $Z_{in}$ is the normalized input impedance; $Z_{0}$ is the free space impedance; and *ε_r_* and *μ_r_* are the complex permittivity and complex permeability, respectively. In general, when the RL < −10 dB, it means that the absorbing material can absorb more than 90% of the incident electromagnetic waves, and when the RL < −20 dB, it means that the absorber will absorb more than 99% of the incident electromagnetic waves [44].

Excellent performance electromagnetic wave absorbing materials are characterized by thin matching thickness, high reflection loss capability, and wide effective absorption bandwidth. From Equations (1) and (2), we can calculate the corresponding material thickness versus reflection loss at 2–18 GHz. It is well known that RLmin value is an important index to evaluate the performance of absorbing materials. Figure 4a–i show the corresponding 3D, 2D and 1D reflection loss plots for all samples in the 2–18 GHz range for frequency and thickness. Comparing the RL values of SiO_2_@NiFe LDH at different matching thicknesses, the S3 sample exhibits excellent electromagnetic wave absorption performance at small thicknesses. The best RL values for S1, S2 and S3 samples are −25.18 dB, −31.76 dB and −53.78 dB, respectively; and the widest effective absorption bandwidths are 4 GHz, 4.3 GHz and 8.24 GHz, respectively. For the S3 sample, a reflection loss of −53.78 dB can be achieved at an ultra-thin thickness of 6.95 mm, and an effective absorption bandwidth of 8.24 GHz (9.76–18 GHz) can be achieved at 9.5 mm, which is significantly better than the matching of most SiO_2_-based core-shell composites’ thickness (Table 1), indicating that it is a promising absorbing material. By adjusting the thickness, the bandwidth of S3 occupies the entire Ku band, which is an extremely excellent performance. The performance gap between S1, S2 and S3 is mainly caused by the difference in lamellar spacing, lamellar thickness and NiFe ratio. The S3 sample has a strong absorption ability and wide effective absorption bandwidth, which is due to that when the NiFe ratio is 3 to 3, the LDH nanosheets have larger interlayer spacing and thicker lamellae, and the increase of the interlayer spacing affects the reflection value. The more frequent scattering is beneficial to the construction of the conductive network and also promotes the polarization loss of the incident electromagnetic wave. The magnetic loss caused by the relatively high Fe content can improve its electromagnetic performance [45]. The basal plane between the NiFe-LDH layers is the active surface, so the size of the interlayer spacing determines the wave absorbing performance, and the small interlayer spacing will restrict the electromagnetic wave from entering the interlayers [46].

In general, the magnetic loss mechanism is caused by eddy current effects, natural resonance and exchange resonance in the gigahertz band. The magnetic loss mechanism is characterized by eddy current, which can be expressed as $\mu^{\prime \prime }{\left(\mu^{\prime }\right)}^{-2}f^{-1}$. The calculated results are shown in Figure 5a. The value of the eddy current on SiO_2_@NiFe LDH fluctuates from 2 to 10 GHz, indicating the eddy current loss is not the main cause of magnetic loss and is consistent with our previous analysis, which can be caused by natural resonances and switching resonances during magnetic loss. In the range of 10–18 GHz, the change of the value of eddy current loss is very small, indicating that the eddy current loss has a greater effect on the attenuation of electromagnetic waves in this frequency range.

The attenuation constant (α) is calculated according to the theoretical formula of the transmission line combined with the measured electromagnetic parameters to reflect the attenuation ability of the absorber [49].
(6)$$\alpha =\left(\sqrt{2}\pi f/c\right)\times \sqrt{\left(\mu^{\prime \prime }\varepsilon^{\prime \prime }-\mu^{\prime }\varepsilon^{\prime }\right)+\sqrt{{\left(\mu^{\prime \prime }\varepsilon^{\prime \prime }-\mu^{\prime }\varepsilon^{\prime }\right)}^{2}+{\left(\mu^{\prime }\varepsilon^{\prime \prime }+\mu^{\prime \prime }\varepsilon^{\prime }\right)}^{2}}}$$
(7)$$c_{0}=\mu^{\prime \prime }{\left(\mu^{\prime }\right)}^{-2}\mu^{\prime \prime }{\left(f\right)}^{-1}=2\pi \mu_{0}d^{2}\delta \;\;$$
where d is the thickness of the absorber and μ is the vacuum permeability. Figure 5b is the correlation curve between the attenuation coefficient and frequency of each sample. With the increase of frequency, the curve changes obviously, but the attenuation system of S3 is more obvious than other samples, which means that S3 has stronger attenuation ability and better absorption performance for incident electromagnetic waves, which is consistent with our previous conclusion.

The impedance-matching characteristic (Z_in_/Z_o_) is another important index to evaluate the electromagnetic wave absorption performance of the wave absorber [50]. For an ideal absorbing material, the reflection value of the incident electromagnetic wave is small or zero is the best. The closer the Z_in_/Z_o_ value is to 1, the easier it is for the incident EMW to pass through the absorber layer. The calculated values of Z_in_/Z_o_ are shown in Figure 5c. The Z_in_/Z_o_ value of the S3 sample is close to 1 and has a smaller fluctuation range. Therefore, the incident electromagnetic wave easily penetrates into the S3 sample and is further attenuated. The optimal impedance matching properties of the S3 sample benefit from the effective tuning of complex permittivity and complex permeability.

Therefore, the functions of each component of the electromagnetic wave absorber are as follows. NiFe LDH, as the main dissipative material, provides excellent conduction losses in core-shell materials. The sheet-like properties of LDH and the SiO_2_ spheres as scaffolds are conducive to the free migration of electrons between sheets, forming a conductive pathway, promoting the effect of conductive penetration, and further increasing the conductive loss. The excellent impedance matching due to the excellent lamellar structure can maximize the entry of incident electromagnetic waves, and the nanoflower-like structure can induce multiple scattering to enhance the absorber interaction. At the permeation threshold, SiO_2_ microspheres act as connectors, which can significantly improve conduction loss and effectively absorb electromagnetic waves.

The electromagnetic wave absorption mechanism of SiO_2_@NiFe LDH is shown in Figure 6. First, NiFe LDH not only has a high specific surface area and abundant inner interface but also has a stable structure, which provides favorable conditions for the maximum attenuation of incident electromagnetic waves inside the absorber in addition to the relatively low electrical conductivity of LDH. It can also provide good impedance matching, which can be obtained by a low dielectric constant, so that more electromagnetic waves can easily enter the material. The electron exchange between ferric ions and divalent nickel ions in NiFe-LDH induces a large number of space charges and polarization charges under the action of alternating electromagnetic fields, both of which increase the dielectric loss capability of the absorber. A large number of heterojunctions between NiFe nanosheets and SiO_2_ matrix generate interfacial polarization, which weakens incident electromagnetic waves and leads to dielectric loss. The nano-sized NiFe nanosheet layered structure can act as a dipole to generate a large number of orientation polarizations and transfer the energy of electromagnetic waves converted into heat energy. Finally, the basal plane between the NiFe-LDH layers is the active surface, which is conducive to the construction of the conductive network and also promotes the polarization loss of the incident electromagnetic wave. 

## 3. Discussion

In conclusion, NiFe LDH was successfully grown in situ on SiO_2_ microspheres by a simple hydrothermal method. The SiO_2_@NiFe LDH nanocomposites with different NiFe element ratios prepared in this paper are expected to provide a new method for the design of novel microwave absorbers. The combined effect of excellent magnetic loss and dielectric loss capabilities, good impedance matching, and the effective combination of excellent spherical and lamellar geometry are the mechanisms for its high-efficiency microwave absorption performance. When the mole ratio of Ni and Fe source is 3:3, the SiO_2_@NiFe LDH-3 obtained has excellent effective absorption bandwidth (9.76 GHz–18.0 GHz) and the composite has the characteristics of high efficiency, no pollution and low cost. In addition, it should be emphasized that in the current reports, LDHs is rarely used for electromagnetic wave absorption. It is undoubtedly a meaningful attempt to make use of its unique structure and good physical and chemical characteristics to prepare a new electromagnetic wave absorber.

## 4. Materials and Methods

### 4.1. Materials

TEOS (Tetraethyl orthosilicate), C_2_H_6_O (ethanol), NH_3_·H_2_O (ammonia), CH_4_N_2_O (urea), Fe (NO_3_)_3_·9H_2_O (Iron nitrate hexahydrate), Ni (NO_3_)_2_·6H_2_O (Nickel nitrate hexahydrate), C_6_H_5_O_7_Na_3_ (trisodium citrate). All chemicals purchased from Aladdin were of analytical purity and used without further purification. 

### 4.2. Synthesis of SiO_2_ Microspheres

Using the improved Stöber method, SiO_2_ microspheres were prepared by catalyzing hydrolysis and agglomeration with ethyl orthosilicate as the raw material and ethanol as the medium. A quantity of 2 mL TEOS was added dropwise to 50 mL of ethanol with continuous stirring. Subsequently, 5 mL NH_3_·H_2_O was slowly added to the above solution. Stirring vigorously for 4 h until the clear solution was converted to a white suspension. The white pellet (SiO_2_ microspheres) was collected by centrifugation, washed repeatedly three times with deionized water and ethanol, and dried at 70 °C for 12 h. 

### 4.3. Synthesis SiO_2_@NiFe LDH Composites

SiO_2_@NiFe LDH composites are prepared by a one-step hydrothermal method. Fe (NO_3_)_3_·9H_2_O, Ni (NO_3_)_2_·6H_2_O urea (12 mmol), and trisodium citrate (0.2 mmol) were dispersed into deionized water (65 mL); SiO_2_ microspheres (80 mg) prepared in the previous step were added and vigorously stirred for 30 min. Subsequently, the mixed solution is transferred to a Teflon-lined stainless autoclave and sealed to heat at 180 °C for 24 h. After the end of the reaction, the samples were washed alternately with ethanol and water three times and dried in vacuo at 60 °C for 12 h. The total metal ion moles were 1.6 mmol, and three samples were prepared by adjusting the molar ratios of Ni^2+^ and Fe^3+^ to 3:1, 3:2, and 3:3, named SiO2@NiFe LDH-1,2,3, abbreviated as S1, S2, S3, respectively. 

### 4.4. Material Characterization

The crystallographic information and chemical compositions of the-prepared nanostructures were established by powder X-ray diffraction (XRD, D/max 2500, Cu Kα), and they were analyzed with the JADE 6.0 software. The magnetic properties were also characterized by a vibrating sample magnetometer (VSM, Lakeshore 7, 304). XPS spectra were acquired on a Physical Electronics ESCA 5600 spectrometer with a monochromatic Al Kα X-ray source (power: 200 W/14 kV) and a multichannel detector (Omni IV). Thermogravimetric analysis (TGA, Mettler Toledo TGA/ DSC3+, in air at a heating rate of 20 °C/min). In the 2.0–18.0 GHz range with vector network analyzer (VNA; Agilent E5071c) to test the electromagnetic parameters of all samples. The samples were mixed with paraffin (mass ratio 4:6) and pressed into rings with an inner diameter of 3.04 mm, outer diameter of 7.00 mm and thickness of 2 mm. The electromagnetic wave absorption properties of the samples were calculated.

## Data Availability

The data presented are contained within the article or Appendix A.

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
