# Peer review of "Core-Shell Structured SiO_2_@NiFe LDH Composite for Broadband Electromagnetic Wave Absorption"

_ijms, 2022, doi:10.3390/ijms24010504_

Round 1
Reviewer 1 Report
The manuscript entitled "Core-shell structured SiO2@NiFe LDH composite for broad-band electromagnetic wave absorption" is interesting. The authors use silica as a substrate and LDH for electromagnetic shielding. The author's work is generally qualified, I think this article can be published in this journal. But I hope the author should seriously consider the following questions before this article is accepted and published:
1. The author should carefully check the quality of the images, especially Fig. 3 (c) and Fig. 3 (f)
2. The author believes that SiO2@NiFe-LDH3 nanosheets are thinner. I hope that the author could provide TEM images of SiO2@NiFe-LDH1/2 nanosheets with a 20-nm scale. It's much more convincing.
3. According to the author's XPS test results, the ratio of Ni and Fe in S3 sample is not 3:3. Can it be considered that the presence of Ni hinders the increase of Fe content in LDH?
4. The author characterized the dielectric constant of the material, if the conductivity of the material can be characterized, I think it will be better.
5. The author believes that SiO2@NiFe LDH-3 achieves the best RL at a thickness of 6.95 mm and the best effective absorption bandwidth at a thickness of 9.5 mm, but the optimal RL and the optimal absorption bandwidth correspond to different thicknesses. I hope the author can explain this situation. (Line 273-276)
6. The author believes that "the performance gap between S1, S2, and S3 is mainly caused by the difference in lamellar spacing, lamellar thickness, and NiFe ratio" (lines 279–280). However, there are no relevant parameters for S1 and S2 in the article. I think the author should provide the relevant data.
Reviewer 2 Report
This manuscript is devoted to the preparation of core-shell structured particles composed of Fe/Ni double layered hydroxides and SiO2 spheres and utilization of such materials as for broad-band electromagnetic wave absorbers.
The material presented in the manuscript is quite strong, the preparation, characterization and electromagnetic wave absorption study are well performed and discussed. The weakest side of the work is poor presentation. The English should be corrected througthout of the mauscript and intensive proof-reading should be performed. My detailed comments are given below.
1) line 58 "To achieve an ideal high-performance SiO2 core-shell material, the following requirements should also be met, that is, there is a good synergy between the SiO 2 core and the shell material so that there is not only a single dielectric loss in the electromagnetic wave loss process[19]." - rephrase please these sentences to make the statement clear!
2) line 97 " Under hydrothermal conditions, urea decomposition of NH 4+ and CO 2 , the hydrolysis process into CO 32- and OH - . " Please check the sentence, something missing!
And further: "Subsequently, metal cations (Ni 2+ and Fe 3+) and OH - generated interaction, in the formation of SiO 2 substrate NiFe LDH, the CO 32- insert NiFe LDH layers structure to achieve a balance between the charge[26]." - What the authors intend to communicate in this sentence?
3) It is impossible to distinguish the element symbols at Figure 2f-i
4) line 123 "...which are the characteristic diffraction peaks of LDH... " - The corresponding references should be provided.
5) line 148 "the weak adsorption peak near 791.36 cm -1 is due to" - It seems that such precision is excessive!
6) line 150 "The weak peak near 1007.78 cm -1 is considered to be the intercalation structure of -OH and CO 3- in hydrotalcite (NiFe LDH). " - What kind of vibrations authors attributes to this peak?
7) Figure 6(c) - What is the reason (from the point of view of chemical composition) causing such a great difference between impedance graphs of LDH3 in comparison to LDH1 and 2 samples?
8) line 374. "Add 2 mL of TEOS dropwise to 50 mL ethanol and stir continuously. " The preparative protocol should be presented using a passive voice. Please, check throughout the Experimental section!
9) line 381 "CH 4 N 2 O (12 mmol), and C 6 H 5 O 7 Na 3 (0.2 mmol)" It is better to provide chemical names instead of formulas here.
10) line 386 "Where the total metal ion concentration is 1.6 mmol, three groups of control experiments are set up by changing the Ni 2+ /Fe 3+ molar ratio. The Molar ratios of Ni 2+ /Fe 3+ were 3:1, 3:2, and 3:3, respectively, and the resulting samples were named SiO 2 @NiFe LDH-1 (S1), SiO 2 @NiFe LDH-2 (S2), and SiO 2 @NiFe LDH-3 (S3)." - Please rewrite this sentence to clarify the preparation. What is "three groups of control experiments"?
Reviewer 3 Report
In the present work, authors reported the synthesis of a novel core-shell structure material SiO2@NiFe LDH by using a one-step hydrothermal 12 method, and then investigated their structural, morphological, and microwave absorption property. Results indicated that composites exhibited broadband effective absorption bandwidth of 8.24 GHz (from 9.76 GHz to 18.0 GHz) and the reflection loss was 53.78 dB with a thickness of 6.95 mm. A series of results and discussion were clearly presented. However, some issues should be addressed.
1, In Introduction section, it was claimed that “In addition, the relatively low conductivity of LDH can provide a good impedance match with the magnetic material composite, so that more electromagnetic waves can easily enter the material, meeting the two advantages of high-quality absorbing materials, therefore, LDH can be considered as a promising electromagnetic wave absorbing materials”. Numerous LDH materials were applied to fabricate absorbers. So why authors chose NiFe LDH as a shell component? Please clarify the significance and advance of NiFe LDH.
2, In morphological section, it was hard to distinguish the core-shell structure of SiO2@NiFe LDH from the TEM images, since there was no distinct hierarchical structure. It is suggested that authors should carry out the map scanning of the single particle in TEM, such as the map scanning of the particle from Figure 2c.
3, It was said that “The weight loss of the S3 sample was 5.5635% at about 150 ℃ and 7.0129% at about 350 ℃. This phenomenon is caused by evaporation of adsorbed water and carbonate anion intercalation transition in NiFe LDH, respectively.”. It is suggested to eliminate the influence of absorbed water before TGA experiment by handling with heat treatment since the first themolysis is meaningless (optional). In addition, the control of the contents of loading NiFe is very important. I am wondering how the loading is controlled and how the content of loading NiFe. I suggest you make a component analysis, such as EDS results.
4, Some key and important research results in absorption field should be mentioned and cited so that we can provide a solid background and progress to the readers, such as Journal of Materials Chemistry C, 2016, 4, 9738; Composites Part A, 2018, 115, 371; Nano-Micro Letters, 2022, 14, 173.
5, The authors have compared with recently reported in the open literature in Table 1. But there is no comparison of absorption peak and content, which are crucial for the real-life application. Please add the data.
6, The introduction of the samples and method used for electromagnetic parameter measurement may be moved into the 2.2 section.
7, The Cole-Cole semicircles were a little disorder, which may confirm the mixed polarization models of SiO2@NiFe LDH rather than one interfacial polarization. Please revised this statement.
8, Why the impedance matching Z values of SiO2@NiFe LDH -3 was weird? Please explain the reason?
Round 2
Reviewer 2 Report
Authors have performed a lot of work to improve the manuscript. In my opinion, the manuscript is now appropriate to be published in International Journal of Molecular Science.
Reviewer 3 Report
All issues were well addressed, and this work can be accepted in the present form.